# IFN γ and the IFN γ Signaling Pathways in Merkel Cell Carcinoma

**DOI:** 10.3390/cancers17152547

**Published:** 2025-08-01

**Authors:** Lina Song, Jinye Guan, Qunmei Zhou, Wenshang Liu, Jürgen C. Becker, Dan Deng

**Affiliations:** 1Department of Dermatology, Shanghai Children’s Medical Center, School of Medicine, Shanghai Jiao Tong University, Shanghai 200127, China; songlina0515@163.com (L.S.); gjy120505@163.com (J.G.); 18270570429@163.com (Q.Z.); sixwinsong@163.com (W.L.); 2German Cancer Consortium (DKTK), Deutsches Krebsforschung Institut, 69120 Heidelberg, Germany

**Keywords:** Merkel cell carcinoma, interferon-gamma (IFN γ), tumor microenvironment, immune evasion, immunotherapy

## Abstract

Interferons are crucial for the host immune response against malignant cells. However, the role of IFN γ and its associated signaling pathways in Merkel cell carcinoma (MCC) remains unclear. This review summarizes recent epigenetic and functional insights into IFN γ in MCC, along with genetic alterations in its associated signaling pathway. We also discuss the potential therapeutic applications of these insights to improve MCC treatment.

## 1. Introduction

Significant progress has been made in understanding the relationships between the host immune system, oncogenesis, and tumor progression. The tumor microenvironment (TME) is increasingly gaining attention, with a growing number of cells and molecules involved in immunotherapy being identified [1]. Interferons (IFNs) are cytokines with pleiotropic biological effects, including the induction of interferon-stimulated gene (ISG) expression, which is essential in immune responses [2]. Among these, type II IFNs act as key mediators of tumor–immune interactions, directly influencing tumor cells in antitumor immune responses. Mechanistically, IFN γ enhances immune recognition of tumor cells by promoting antigen presentation and activating the major histocompatibility complex (MHC) class I pathway, highlighting its critical role in early antigen presentation [3]. Notably, Grasso et al. suggested that antitumor T-cell responses activated by the IFN γ signaling pathway are the primary effectors in melanoma immune checkpoint inhibitor (ICI) therapy [4].

The TME is composed of tumor cells, stromal cells, immune cells, and the extracellular matrix containing various cytokines, all of which are potentially involved in tumor progression and immune evasion [5]. Enhancing the immune system‘s ability to recognize and eleminate tumor cells is currently the most effective method for controlling tumor growth, with IFN γ playing a particularly important role [6]. This cytokine regulates various immune functions through IFN γ, including the activation of cytotoxic T lymphocytes (CTLs), natural killer (NK) cells, and macrophages, which are crucial for tumor immune responses [6].

Merkel cell carcinoma (MCC) is an immunogenic, but still highly aggressive skin cancer characterized by a high recurrence rate associated with pronounced mortality [7]. In recent years, ICI therapy has shown promising results in MCC patients [8]. However, primary and secondary resistance are common, and at least half of advanced MCC patients do not benefit from ICI therapy. One contributing factor is the downregulation of MHC I expression, which can be restored by histone deacetylase (HDAC) inhibitors [9]. Furthermore, emerging studies indicate that loss-of-function mutations and genomic alterations in antigen presentation and IFN signaling pathways contribute to tumor resistance to ICI [10,11]. Recent transcriptomic studies provide additional insights into the role of IFN γ in immune escape mechanisms. Tatjana et al. employed single-molecule RNA sequencing to investigate how IFN γ influences gene expression in Merkel cell polyomavirus (MCPyV)-positive MCC cell lines. Their results revealed that IFN γ treatment significantly upregulates a set of genes involved in immune evasion, including members of the suppressor of cytokine signaling and interferon regulatory factor families, suggesting the dual role of IFN γ in both enhancing immune activation and promoting tumor immune escape [12].

Given the pivotal role of IFN γ signaling in anti-tumor responses, impaired IFN γ signaling is increasingly acknowledged as a mechanism underlying both cancer immune escape and resistance to ICI [13]. Therefore, it is crucial to investigate the function and integrity of the IFN γ signaling pathway in MCC for a deeper understanding of its initiation and progression, as well as for the development of novel therapeutic strategies for MCC. Notably, recent preclinical efforts have explored vaccine-based immunotherapy approaches. For example, Kerstin et al. demonstrated that dendritic cells engineered to express the truncated large T antigen of MCPyV, combined with constitutive nuclear factor kappa B (NF-κB) activation to enhance immunogenicity, successfully elicited virus-specific T-cell responses in healthy donors and some MCC patients [14]. These findings highlight the therapeutic potential of dendritic-cell-based vaccination in virus-driven cancers such as MCC.

This review aims to offer a comprehensive examination of IFN γ and its associated signaling pathways in the context of MCC, highlighting their intricate role in immune regulation and their therapeutic potential. Furthermore, we summarize the exploration of the role of the IFN γ signaling pathway in MCC and address the challenges and future prospects of IFN-γ-based immunotherapy, focusing on how these insights can be translated into viable therapeutic options.

## 2. MCC and IFN γ

MCC is an immunogenic but highly aggressive skin cancer with neuroendocrine differentiation [15]. It is marked by rapid growth and typically presents as firm, red, or purple nodules on sun-exposed areas such as the head, neck, and limbs, primarily in elderly individuals. MCC has a high propensity for metastasis [16]. Although relatively rare with a low incidence rate, the incidence of MCC has been increasing in recent decades, particularly in the aging populations. Notably, MCC is more common among individuals with compromised immune function, in whom it also exhibits more aggressive behavior, thus confirming the relevance of immune surveillance for MCC. The pathogenesis of MCC is complex, involving various factors, i.e., UV-associated gene mutations or clonal integration of MCPyV during viral oncogenesis (Figure 1) [17,18].

MCPyV is a small, non-enveloped, double-stranded DNA virus that contains a genome with three regions: a noncoding control region (NCCR), an early coding region (ECR), and a late coding region (LCR). In MCC pathogenesis, MCPyV plays a pivotal role through its two early genes: large transforming (LT) and small transforming antigens (sT) [19]. The LT antigen interacts with the retinoblastoma protein (Rb) via its N-terminal LXCXE motif, which releases Rb‘s inhibition of the E2F transcription factor, thereby promoting the progression of the cell cycle [20]. The sT antigen, through its unique C-terminal domain, binds to various host proteins, including cyclin-dependent kinase 20 (CDC20), F-box and WD-repeat-domain-containing protein 7 (FBXW7), and protein phosphatase 2A (PP2A) [21]. This binding of sT to CDC20 enhances the phosphorylation of the eukaryotic translation initiation factor 4E-binding protein 1 (4E-BP1), leading to the overproduction of cellular mRNA and promoting cell proliferation [22]. Additionally, sT forms complexes with MYCL and EP400 proteins, activating MDM2 and MDM4 expression while inhibiting p53 activity, thus facilitating cell transformation [23]. The integration of MCPyV DNA into the host cell genome is a critical step in MCC development, resulting in the persistent expression of T antigens that drive malignant transformation [24].

The treatment approaches for MCC include surgery, radiation therapy, chemotherapy and immunotherapy [25]. Surgical resection, either alone or combined with adjuvant radiation therapy, is the standard first-line treatment for localized tumors without metastatic spread [25]. In the advanced setting, it is important to note that MCC shows a high response rate to cytotoxic agents even if used as monotherapy; unfortunately, however, MCC rapidly develops secondary resistances, significantly limiting the effectiveness of conventional chemotherapy regimens [26]. While early localized MCC can often be managed with surgery and adjuvant radiation, approximately 30–40% of patients develop metastatic or locally advanced tumors that are unresectable or refractory to conventional therapies [27], highlighting the urgent need for novel therapeutic strategies.

The discovery of MCPyV integration in MCCs and the identification of ultraviolet-induced mutagenesis in virus-negative tumors provided critical insights into tumor immunogenicity [28]. These findings, coupled with the success of immune checkpoint inhibitors (ICI) in other immunogenic cancers, positioned MCC as a prime candidate for PD-1/PD-L1 blockade. Landmark clinical trials, including KEYNOTE-017 [29] and JAVELIN Merkel 200 [30], demonstrated the efficacy of pembrolizumab and avelumab, respectively, with objective response rates (ORR) of 30–56%, durable complete remissions, and two-year OS rates exceeding 60%. These breakthroughs established ICI as the cornerstone of first-line therapy for advanced MCC, as endorsed by current NCCN (2024) and ESMO guidelines. Thus, the advent of immunotherapy, particularly ICI targeting the PD-1/PD-L1 axis (e.g., pembrolizumab and avelumab), has transformed the treatment landscape for advanced-stage MCC [31,32]. These agents have shown significant and lasting antitumor effects, establishing them as key therapies for metastatic or recurrent MCC. Ongoing research is focused on evaluating the role of immunotherapy in neoadjuvant and adjuvant settings and optimizing combination strategies to improve therapeutic outcomes and delay resistance development [26,33].

Despite significant advancements in immunotherapy, intrinsic or acquired resistance to ICI affects nearly half of patients, and subsets with immunosuppression or MCPyV-negative status exhibit attenuated responses, highlighting the urgent need for further exploration of novel targeted treatment strategies [34]. In recent years, emerging strategies to overcome these barriers—including combination therapies, novel immune targets, and personalized approaches—are now at the forefront of clinical investigation [33]. As a member of the type II interferon family, IFN γ is primarily secreted by immune cells, such as CD4^+^ Th1 cells, CD8^+^ cytotoxic T cells, NK cells, and some macrophages [35]. As a key mediator of the immune response, IFN γ plays a crucial role in enhancing the cytotoxic activity of NK cells and CTLs, which is essential for recognizing and eliminating tumor cells, including those in MCC [36,37]. Additionally, IFN γ regulates macrophage polarization, promoting their shift toward the proinflammatory M1 phenotype, which is associated with increased antigen presentation and the activation of antitumor immunity [38]. This process improves the immune system’s ability to detect and target tumor cells, which is critical for controlling cancer growth. Moreover, IFN γ upregulates the expression of MHC class I and II molecules on dendritic cells, leading to the induction of effective cytotoxic T-cell responses and further enhancing the immune system’s ability to recognize tumor-associated antigens [39]. Figure 2 illustrates the sources of IFN γ and its role in activating the antitumor immune response within the TME.

Approximately 80% of MCC cases harbor a clonally integrated truncated MCPyV genome coding for sT and a truncated form of LTA expressed from the viral promoter [40,41]. IFN γ, a type II interferon primarily produced by activated T cells and NKcells, exerts potent antiviral effects not only by directly suppressing viral replication but predominantly through immune-mediated cytotoxicity against virus-infected cells [42]. Emerging studies suggest that IFN γ may exhibit antitumor activity against MCC through both direct immunomodulatory effects and indirect mechanisms targeting the tumor microenvironment (TME) [43]. Specifically, IFN γ enhances antigen presentation by upregulating MHC class I/II molecules on tumor cells and dendritic cells (DCs), thereby promoting cytotoxic T lymphocyte (CTL)-mediated tumor cell killing [44]. In MCC, which is strongly associated with MCPyV, IFN γ may also suppress viral oncoprotein-driven tumorigenesis by restoring p53 pathway activity [45,46]. Preclinical models further indicate that IFN γ induces tumor vascular regression via endothelial cell-specific signaling, leading to ischemia-mediated tumor necrosis—a mechanism analogous to its effects in other solid tumors [47]. Additionally, IFN-γ-mediated activation of Janus Kinase/Signal Transducer and Activator of Transcription (JAK/STAT) pathways enhances the recruitment of immune cells (e.g., NK cells and macrophages) to the TME [48], which may synergize with ICIs to overcome resistance in refractory MCC. These findings position IFN γ as a promising therapeutic adjunct in MCC management, particularly for tumors that are resistant to conventional therapies [49]. Recent research has shown that neoantigen-specific CD4^+^ T cells in MCC tumors produce substantial amounts of IFN γ when activated by tumor-derived neoantigen peptides, further supporting the importance of IFN γ in antitumor immune responses [50].

Indeed, the molecular mechanisms underlying IFN γ signaling remain complex and are not fully elucidated in MCC. The canonical IFN γ signaling pathway, which involves the JAK-STAT pathway, is likely to be involved. With the binding of IFN γ to its receptor on MCC cells, JAK1 and JAK2 are activated, leading to the phosphorylation of signal transducer and activator of transcription 1 (STAT1). The phosphorylated STAT1 then translocates to the nucleus and regulates the transcription of genes responsive to IFN γ [38]. However, additional mechanisms specific to MCC may exist. For example, the stimulator of interferon genes (STING) pathway, which senses cytoplasmic DNA, may interact with IFN γ signaling [51]. In some cases, STING agonists have been shown to enhance antitumor immunity in MCC by indirectly signaling through immune and stromal cells in the TME, potentially modulating IFN γ-related responses [39]. Additionally, MCC cells may exhibit unique epigenetic modifications that influence IFN γ signaling. For instance, recent studies have demonstrated that histone demethylases such as KDM7A (also known as JHDM10) can modulate the JAK2/STAT1 pathway by altering chromatin structure and affecting protein methylation status [52]. Given the functional parallels in immune evasion strategies between viruses and tumors, it is plausible that similar epigenetic mechanisms may operate in MCC, contributing to immune escape. Understanding these regulatory processes could provide new insights into the development of targeted therapies that modulate IFN γ signaling in MCC.

Despite its potent antitumor effects, the role of IFN γ in MCC is intricate and may be influenced by the TME, which often contains immunosuppressive factors that limit the efficacy of IFN γ [53]. Like many other cancers, the TME of MCC may harbor various immunosuppressive components, such as regulatory T cells (Tregs), myeloid-derived suppressor cells (MDSCs), and numerous soluble immune checkpoints, all of which collectively suppress immune responses and hinder effective tumor elimination [54]. These immunosuppressive factors can diminish the action of IFN γ, restricting its ability to activate immune cells or induce tumor cell apoptosis. To overcome these challenges, strategies to enhance the efficacy of IFN γ in MCC are being explored. Combining IFN γ therapy with ICI or other immunotherapies may help counteract the immune evasion mechanisms present in the TME [55].

## 3. The Role of IFN γ in the MCC Microenvironment: Antitumor and Immune Evasion Effects

The role of IFN γ in the MCC microenvironment is multifaceted, contributing both to significant antitumor effects and to immune evasion and drug resistance mechanisms. Figure 3 illustrates this dual role of IFN γ.

Specifically, IFN γ exerts anti-tumor effects by inducing tumor cell apoptosis, enhancing antigen presentation, and promoting the cytotoxic activity of immune cells against tumors [56]. Conversely, it facilitates immune evasion by upregulating PD-L1 expression on tumor cells, inducing the secretion of immunosuppressive factors, and promoting EMT, which aids tumor progression [57]. This figure emphasizes the complex role of IFN γ in the TME, highlighting its capacity to enhance both antitumor immunity and immune evasion.

In MCC, IFN γ has been shown to upregulate the expression of cytidine deaminase in the apolipoprotein B mRNA-editing catalytic polypeptide-like (APOBEC) family, particularly APOBEC3B and APOBEC3G, in MCPyV-positive cell lines [58]. This induction is associated with cytosine mutations that preferentially occur in TpC dinucleotide contexts, a mutation signature consistent with APOBEC-mediated activity. Studies have shown that IFN γ plays a central role in shaping the MCPyV genome through the induction of APOBEC family cytidine deaminases, such as APOBEC3A, 3B, and 3G. Specifically, IFN γ treatment significantly upregulates APOBEC3B and APOBEC3G expression in MCPyV-positive MCC cell lines, leading to cytosine mutations with a strong TpC dinucleotide preference within the viral genome [58]. Although classical kataegis—defined as localized hypermutation clusters—was not explicitly reported in this study, the mutation pattern is consistent with APOBEC-mediated mutagenesis, which is known to underlie kataegis in other cancer types. Thus, IFN-γ-induced APOBEC expression may contribute to viral genome editing and the accumulation of tumor-specific mutations in MCPyV-positive MCC. In addition, IFN γ has been shown to inhibit tumor cell proliferation and induce apoptosis in MCC cells [59,60,61], potentially through mechanisms such as the modulation of oxidative stress [12,62] and the activation of downstream pathways including the canonical JAK–STAT axis, as previously described.

IFN γ can also enhance immune cell function by regulating immune cell infiltration in the TME. For example, IFN γ can promote the maturation and antigen-presenting capacity of dendritic cells (DCs) [63]. Other studies have reported that IFN γ can promote the transmission of inflammatory signals, attracting more immune cells to the tumor site. In the IFN γRKO tumor model, IFN γ accumulation enhances inflammatory signaling within the tumor, particularly increasing the recruitment of monocytes and inflammatory macrophages [64]. Thus, even IFN-γ-insensitive tumor cells can reshape the tumor immune microenvironment by accumulating IFN γ, especially by promoting the recruitment and differentiation of monocytes. Monocytes synergize with CD8^+^ T cells around tumor blood vessels, boosting CD8^+^ T cell activity and improving their ability to recognize and kill tumor cells [64].

Upon binding to its receptor on tumor cells, IFN γ activates JAK1 and JAK2, which in turn phosphorylate STAT1. This induces STAT1 dimerization and nuclear translocation, where it binds to the PD-L1 gene promoter, enhancing PD-L1 transcription and expression. The figure illustrates how IFN γ upregulates PD-L1 in tumor cells, facilitating immune evasion and contributing to resistance against immune-mediated destruction. 

Despite its potent antitumor functions, IFN γ can paradoxically promote immune evasion and tumor progression through several mechanisms. One such mechanism involves the epigenetic suppression of the expression of the major histocompatibility complex class II [65]. In colorectal cancer cells, for instance, promoter methylation of the class II transactivator (CIITA) gene impairs IFN-γ-induced CIITA expression, thereby blocking MHC class II upregulation and reducing CD4^+^T cell recognition [65]. IFN γ also modulates the tumor immune microenvironment by inducing the expression of immune checkpoint and immunosuppressive molecules. It promotes programmed death-ligand 1 (PD-L1) expression via the Janus Kinase/signal transducer and activator of transcription 1 (JAK–STAT1) pathway in both tumor and immune cells, contributing to immune evasion and reduced cytotoxic T cell activity [57,66]. As illustrated in Figure 4, defects in this pathway may result in resistance to ICI therapy. In addition, IFN γ upregulates indoleamine 2,3-dioxygenase 1 (IDO1), an immunoregulatory enzyme involved in tryptophan catabolism. Mechanistically, IFN γ activates STAT1, which cooperates with CREB-binding protein (CBP)/p300 to remodel chromatin and enhance accessibility of the IDO1 promoter. This facilitates the transcriptional activation of IDO1, resulting in elevated kynurenine levels that suppress CD8^+^ T cell function and promote regulatory T cell expansion [67]. Furthermore, IFN γ signaling has been implicated in the induction of epithelial-to-mesenchymal transition (EMT), thereby increasing tumor cell motility and metastatic potential. This process is believed to be mediated through the sustained activation of STAT1 and the downstream transcription factor interferon regulatory factor 1 (IRF1), which can modulate EMT-related gene networks including SNAIL, ZEB1, and TWIST. Additionally, IFN γ may synergize with transforming growth factor-beta (TGF-β) signaling to reinforce EMT programs, particularly in the presence of chronic inflammation [57].

While STAT1 is the canonical IFN-γ signaling mediator, other STAT members such as STAT3 and STAT5 also influence immune responses. STAT3, often aberrantly activated in tumors, drives immunosuppression by inducing IL-10 and TGF-β and expanding MDSCs. It can antagonize STAT1 by competing for upstream kinases and DNA binding and may be indirectly activated via cytokine crosstalk, promoting tumor progression [68,69]. STAT5, primarily activated by γ-chain cytokines such as IL-2 and IL-7, regulates T cell differentiation and survival, contributing to regulatory T cell expansion and immune evasion [70]. These findings highlight the complexity of STAT signaling crosstalk and suggest that targeting multiple STAT pathways may improve IFN-γ-based therapies, especially in MCC [71].

In summary, the role of IFN γ in the MCC microenvironment is inherently dualistic, with its antitumor effects intertwined with its capacity to promote immune evasion and drug resistance. On one hand, IFN γ significantly boosts the antitumor immune response by inducing apoptosis, enhancing immune cell activity, promoting antigen presentation, and upregulating the expression of immune-related genes, thus inhibiting tumor cell growth and metastasis. On the other hand, IFN γ may also drive tumor cells to express immunosuppressive molecules (such as PD-L1) and recruit immunosuppressive cells (such as MDSCs and TAMs), thereby diminishing the immune system’s ability to target tumor cells and facilitating immune evasion and resistance to treatment. This dual role emphasizes the need to consider both the beneficial and detrimental effects of IFN γ in the treatment of MCC, in order to optimize immunotherapy strategies.

## 4. IFN γ and Immunotherapy for MCC

Since 2018, ICI agents targeting PD-1 and PD-L1 have been approved for the treatment of advanced MCC, demonstrating remarkable clinical success [29,72]. Approximately 50% of MCC patients respond to ICI therapy, which has become the standard treatment for metastatic MCC [73]. However, the other half of patients still exhibit resistance to ICI therapy [8,9].

Resistance to anti-PD1 and anti-CTLA-4 therapies has recently been linked to IFN γ resistance, which protects tumors from cytokine-induced apoptosis, cell cycle arrest, and sustained IFN γ signaling that upregulates ligands for inhibitory T-cell receptors [74,75,76]. Additionally, previous studies have shown that defects in the IFN γ signaling pathway are associated with resistance to ICI therapy [4]. In melanoma, the IFN γ signaling signature correlates closely with clinical responses to immunotherapy, acting as a primary driver of either response or resistance to treatment [4]. Loss-of-function mutations and genomic alterations in the IFN γ signaling pathway lead to immune evasion, causing tumor resistance to ICI therapy [74,75,77]. However, in most tumors, such as colorectal cancer as well as MCC, genetic mutations in the IFN γ signaling pathway are either rare or have not been previously described [78].

Given these observations, combining IFN γ with checkpoint inhibitors may provide a synergistic effect, as IFN γ enhances tumor immunogenicity and may augment the efficacy of checkpoint blockade [79]. Although IFN γ is not currently approved as a standard cancer therapy, several preclinical studies and early-phase clinical trials have explored its potential. For instance, IFN γ has been investigated in combination with anti-PD-1 or anti-CTLA-4 therapies in melanoma, renal cell carcinoma, and ovarian cancer, where it showed enhanced T cell infiltration and improved antitumor responses [80,81,82]. These findings suggest that IFN γ could serve as an immunomodulatory adjuvant in combination immunotherapy strategies. Additionally, IFN-γ-based therapies may synergize with established treatments such as chemotherapy or radiotherapy, both of which can induce immunogenic cell death and promote the release of tumor-associated antigens recognized by the immune system [83,84]. By enhancing antigen presentation and T cell activation, these conventional treatments may create a more favorable immune contexture in which IFN γ can exert its imuunomodulatory effects. Therefore, elucidating the mechanisms that compromise the integrity of IFN γ signaling in MCCs is an attractive area for further research.

## 5. Enhancing IFN γ Therapy for MCCs: Overcoming Immune Evasion and Resistance

Despite the substantial potential of IFN γ in MCC treatment, its clinical application faces significant challenges, primarily due to immune evasion, which can markedly reduce the efficacy of IFN γ [85]. The immunosuppressive effects induced by MCPyV play a crucial role in regulating immune responses, impairing the ability of immune cells to effectively target and eliminate tumor cells [86]. MCPyV-encoded oncoproteins, particularly the large T (LT) and small T (sT) antigens, can downregulate immune recognition by interfering with MHC class I expression, suppressing antigen presentation, and modulating interferon signaling pathways [7]. These viral proteins may also alter the tumor microenvironment by recruiting immunosuppressive cells such as regulatory T cells and myeloid-derived suppressor cells, thereby facilitating immune evasion [87,88]. This suppressive milieu impairs the activation and function of key cytotoxic immune cells, such as CD8^+^ T cells and natural killer (NK) cells, thereby limiting the antitumor efficacy of IFN γ.

Moreover, despite its efficacy in some settings, such as the prevention of severe infections in chronic granulomatous disease and the treatment of malignant osteopetrosis and some hematologic malignancies, the clinical use of IFN γ is often associated with side effects, such as flu-like symptoms, fatigue, and hematologic toxicity [89,90]. These adverse effects, common to many immune-based therapies, can significantly reduce patient tolerance and may limit the overall benefits of IFN γ in clinical practice. Additionally, prolonged IFN γ therapy may lead to the development of resistance mechanisms that undermine its effectiveness in treating MCC. Notably, mutations in the IFN γ receptor (IFNGR1 and IFNGR2) and downstream signaling molecules, such as STAT1, have been identified as key factors in resistance, as they impair the signaling cascade necessary for IFN-γ-mediated tumor cell death [91]. Such mutations can result in immune evasion, with tumor cells becoming unresponsive to IFN γ, thereby limiting its clinical success.

Personalized treatment strategies will also be vital for optimizing IFN γ therapy. Identifying biomarkers to predict patient responses to IFN γ and other immunotherapies, along with genomic and proteomic analyses to reveal molecular characteristics of immune resistance, will enable clinicians to tailor treatment plans based on individual patient profiles. One study used single-molecule RNA sequencing to analyze the transcriptomic effects of IFN γ on MCC cell lines (WaGa, MKL-1, and MKL-2), reporting significant changes in gene expression following IFN γ treatment [12]. In particular, the study observed the upregulation of multiple immune evasion-related genes, including PD-L1 (CD274), HLA class I molecules (HLA-A/B/C), and β2-microglobulin (B2M), suggesting that IFN γ may influence its antitumor effects by modulating tumor cell immune escape mechanisms [12].

Clinical trials related to IFN γ modulation in MCC are still relatively limited. However, some studies in other cancers have provided insights. For instance, in patients with solid tumors treated with low-dose DNA-demethylating agents, an increase in IFN γ^+^ T cells was correlated with improved therapeutic responses and survival. This suggests that strategies aimed at enhancing IFN-γ-related immune responses could be beneficial in MCC treatment [71]. In recent years, the combination of IFN γ with other immunotherapeutic agents has garnered increasing attention. For example, preclinical studies have shown that combining IFN γ with anti-PD-1/PD-L1 antibodies results in synergistic effects, such as enhanced T cell infiltration, upregulation of MHC class I molecules, increased expression of chemokines such as CXCL10, and improved dendritic cell activation, leading to more robust antitumor immunity [92,93]. This combination therapy holds promise for improving treatment outcomes in MCC patients, particularly those with advanced or metastatic disease. Furthermore, combining IFN γ with other immune modulators, such as other ICIs (e.g., LAG3 or TIGIT directed antibodies), cytokine therapies, or oncolytic viruses, may help overcome the limitations of monotherapy [55,94]. These strategies are expected to enhance therapeutic outcomes by amplifying immune responses and overcoming immune resistance mechanisms in MCC. To further improve IFN-γ-based immunotherapy, targeting key immune evasion nodes within the IFN γ pathway has emerged as a promising strategy. For instance, IDO1 inhibitors such as epacadostat or linrodostat may counteract IFN-γ-induced immunosuppression by reducing kynurenine-mediated T cell dysfunction [95]. Epigenetic modulators, including DNA methyltransferase inhibitors and HDAC inhibitors, may restore MHC class II expression by reversing CIITA promoter silencing, thereby enhancing tumor immunogenicity and responsiveness to IFN γ [96]. Moreover, IFN-γ-driven signaling often interacts with the STAT3 pathway, which suppresses immune activation via IL-10 and TGF-β and recruits MDSCs. Small-molecule STAT3 inhibitors, such as napabucasin, are being explored to reinforce STAT1-mediated antitumor immunity and reduce immunosuppressive feedback [97]. These therapeutic targets align with the immunosuppressive mechanisms illustrated in Figure 4, particularly those involving IFN-γ-induced PD-L1 upregulation. By combining IFN γ with pathway-specific inhibitors or epigenetic drugs, it may be possible to reprogram the tumor microenvironment, alleviate immune suppression, and enhance the therapeutic efficacy of IFN-γ-based regimens in MCC.

Although IFN γ therapy has only been approved for chronic granulomatous disease and severe malignant osteopetrosis, several early-phase and translational studies have explored its potential role in treating MCC [93]. Notably, pilot efforts combing IFN γ with immune checkpoint blockade or adoptive T-cell therapy have demonstrated acceptable safety profiles and signs of clinical activity, even in patients refractory to standard immunotherapy [98]. These exploratory investigations [99] lay important groundwork for future clinical trials and support the rationale for incorporating IFN-γ-based strategies into MCC treatment paradigms.

## 6. Conclusions

IFN γ has significant potential for treating MCC. As a key mediator of the immune response, IFN γ plays an essential role in enhancing antitumor immunity by promoting cytotoxic activity, antigen presentation, and the activation of immune cells such as NK cells and T lymphocytes. However, its therapeutic efficacy in MCC is hampered by immune evasion mechanisms, immunosuppressive factors in the TME, and resistance to IFN γ signaling. These challenges could be further exacerbated by genetic mutations within the IFN γ signaling pathway. Future research should focus on overcoming these obstacles by targeting immune evasion mechanisms, enhancing our understanding of the TME, and identifying biomarkers for personalized treatment strategies. Combination therapies involving IFN γ and other immune modulators, such as checkpoint inhibitors, may help counteract the suppressive effects of the TME and improve treatment outcomes. Personalized approaches based on genomics and proteomics may also increase the likelihood of successful treatment. Ultimately, continued research into IFN-γ-based therapies is vital to improving clinical outcomes of MCC patients, offering hope for more effective and personalized treatments.

## Figures and Tables

**Figure 1 cancers-17-02547-f001:**
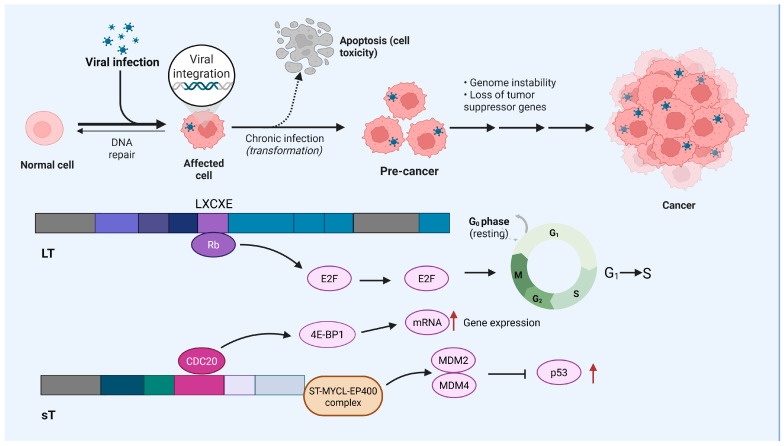
Schematic diagram of the pathogenic mechanism of MCC caused by MCPyV infection.

**Figure 2 cancers-17-02547-f002:**
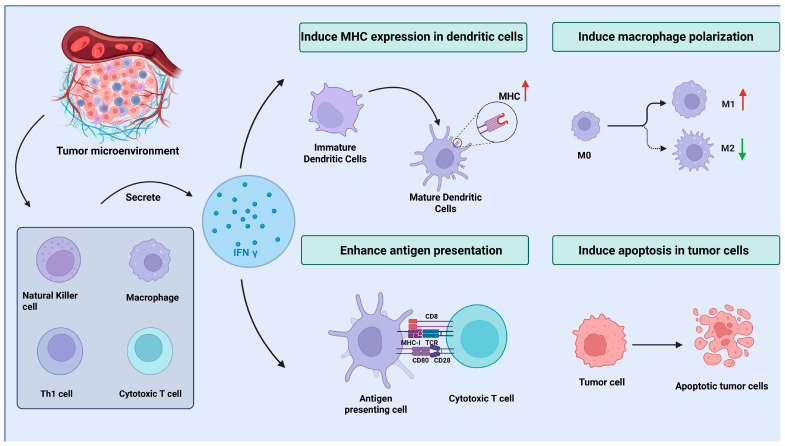
The sources of IFN γ and its role in activating the antitumor immune response within the TME. Created using BioRender.com.

**Figure 3 cancers-17-02547-f003:**
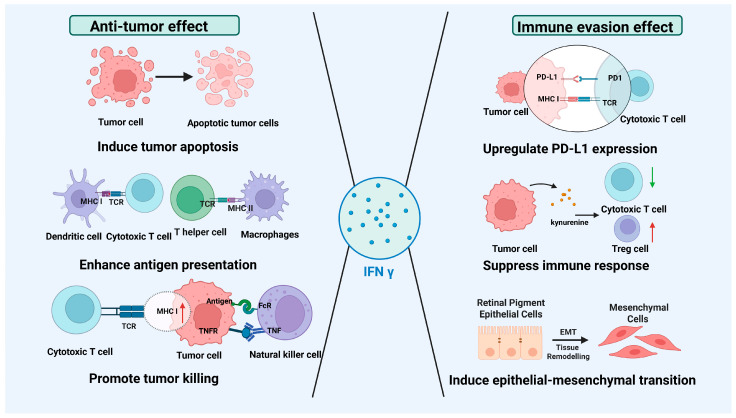
Dual effects of IFN γ in the TME: anti-tumor and immune evasion effects. Created using BioRender.com.

**Figure 4 cancers-17-02547-f004:**
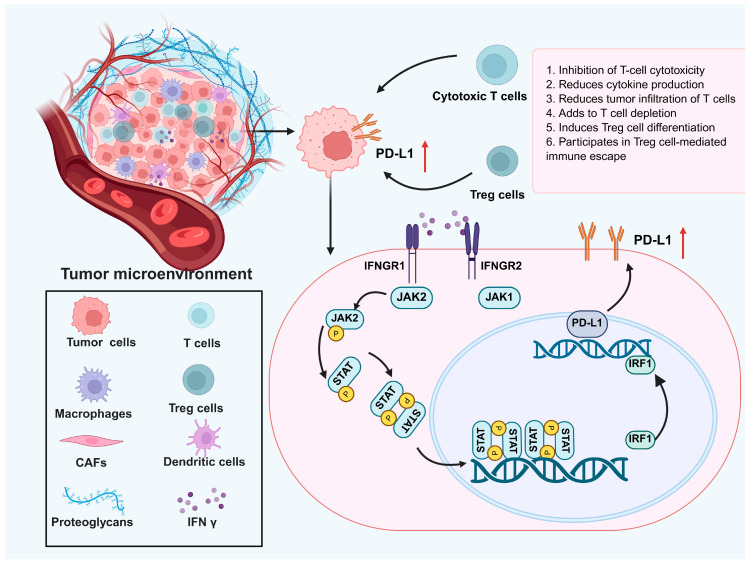
Mechanism of IFN-γ-induced PD-L1 upregulation and its impact on immune evasion in the TME.

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
