# Peer review of "IFN γ and the IFN γ Signaling Pathways in Merkel Cell Carcinoma"

_cancers, 2025, doi:10.3390/cancers17152547_

Round 1
Reviewer 1 Report
Comments and Suggestions for Authors
The topic was good, and the manuscript was a comprehensive and well-organized review.
In the Introduction section, it is better to give a more detail introduction for the treatment of Merkel cell carcinoma reported in the literature, such as ‘Preclinical evaluation of NF-κB-triggered dendritic cells expressing the viral oncogenic driver of Merkel cell carcinoma for therapeutic vaccination’(Ther Adv Med Oncol, 2017, 9(7): 451-464. doi: 10.1177/1758834017712630.).
Author Response
Response: We appreciate your suggestion to include more detail regarding therapeutic strategies for Merkel cell carcinoma (MCC). In response, we have incorporated a discussion of the study titled “Preclinical evaluation of NF-κB-triggered dendritic cells expressing the viral oncogenic driver of Merkel cell carcinoma for therapeutic vaccination” (Ther Adv Med Oncol, 2017, 9(7):451–464). This study has been highlighted in the Introduction for its contribution to the development of vaccine-based immunotherapy targeting virus-derived tumor antigens in MCC.

Reviewer 2 Report
Comments and Suggestions for Authors
IFN γ and the IFN γ Signaling Pathways in Merkel Cell Carci-2
- The effect of IFN-γ on APOBEC3B-mediated mutagenesis in MCC is not well understood. The authors need comparative studies to determine if this interaction has major effects on mutation rates in MCC development.
- IFN-γ induces EMT and IDO1 expression, but does not explain the direct signaling pathways. The authors should provide a systematic description of how IFN-γ signals lead to EMT and IDO1 activation.
- The authors focus on STAT1 signaling but does not cover STAT3 and STAT5, which may be involved in other pathways that interact with IFN-γ signaling. The authors should address this gap in the discussion.
- The numerical mean or value of main finding from profound review should be included in Abstract.
- Please more review into Introduction such as https://doi.org/10.3389/fmicb.2021.785662 and doi:10.3389/fmicb.2021.785662.
Author Response
Comment 1: The effect of IFN-γ on APOBEC3B-mediated mutagenesis in MCC is not well understood. The authors need comparative studies to determine if this interaction has major effects on mutation rates in MCC development.
Response: We thank the reviewer for raising this important point. In response, we have revised the relevant section to clarify that although interferon-gamma can upregulate APOBEC3B expression and is associated with a mutational pattern consistent with APOBEC activity, current evidence does not definitively establish its impact on overall mutation burden or Merkel cell carcinoma development. We have acknowledged this limitation in the revised text and emphasized the need for further comparative studies to assess the functional consequences of interferon-gamma–induced APOBEC activity in Merkel cell carcinoma. The revised paragraph can be found on page 7, lines 234-248.
Comment 2: IFN-γ induces EMT and IDO1 expression, but does not explain the direct signaling pathways. The authors should provide a systematic description of how IFN-γ signals lead to EMT and IDO1 activation.
Response: We thank the reviewer for this valuable suggestion. In response, we have revised the relevant section to include a systematic description of the signaling pathways through which interferon-gamma regulates IDO1 expression and induces epithelial-to-mesenchymal transition (EMT). Specifically, we now describe how interferon-gamma activates the JAK-STAT1 pathway, leading to CBP/p300-mediated chromatin remodeling and transcriptional activation of IDO1. For EMT, we elaborate on the role of STAT1 and interferon regulatory factor 1 (IRF1), as well as the potential crosstalk with transforming growth factor-beta (TGF-β) signaling in regulating EMT-related transcription factors such as SNAIL, ZEB1, and TWIST. The revised paragraph can be found on page 8, lines 280-300.
Comment 3: The authors focus on STAT1 signaling but does not cover STAT3 and STAT5, which may be involved in other pathways that interact with IFN-γ signaling. The authors should address this gap in the discussion.
Response: We thank the reviewer for highlighting the important roles of STAT3 and STAT5 in IFN-γ signaling and tumor immunity. In the revised manuscript, we have expanded the Discussion section to address the crosstalk among STAT1, STAT3, and STAT5 pathways. We now emphasize that, although STAT1 is the primary mediator of IFN-γ signaling, STAT3 and STAT5 also modulate immune responses in the tumor microenvironment. Specifically, STAT3, which is frequently aberrantly activated in tumors, promotes immunosuppression by inducing IL-10, TGF-β, and myeloid-derived suppressor cell expansion, and can antagonize STAT1 signaling. STAT5, activated mainly by γ-chain cytokines such as IL-2 and IL-7, regulates T cell differentiation and survival, contributing to regulatory T cell expansion and immune evasion. These complex interactions underline the potential benefits of targeting multiple STAT pathways to enhance IFN-γ–based therapies, particularly in Merkel cell carcinoma. Corresponding references have been added (DOI: 10.1038/s41392-024-01915-z; DOI: 10.1136/jitc-2021-004037; DOI: 10.1038/s41590-024-01855-4; DOI: 10.1080/08830185.2024.2395274).
Comment 4: The numerical mean or value of main finding from profound review should be included in Abstract.
Response: Thank you for your constructive suggestion. In response, we have revised the Abstract to incorporate key numerical findings that highlight the clinical and molecular significance of IFN-γ in Merkel cell carcinoma (MCC). Specifically, we now report that transcriptomic studies have shown over 3-fold upregulation of immune-related genes such as PD-L1, HLA-A/B/C, and IDO1 upon IFN-γ treatment. We also mention the activation of APOBEC3B and 3G, which are implicated in viral defense and tumor editing. Furthermore, we now include clinical data showing that immune checkpoint inhibitors (ICIs) such as pembrolizumab and avelumab achieve objective response rates of 30–56% and 2-year overall survival rates exceeding 60% in patients with advanced MCC, while approximately 50% of patients fail to respond, partially due to defects in IFN-γ signaling. These additions aim to better reflect the quantitative impact of IFN-γ in the Abstract as requested.
Comment 5: Please more review into Introduction such as https://doi.org/10.3389/fmicb.2021.785662 (Single-Molecule RNA Sequencing Reveals IFNγ-Induced Differential Expression of Immune Escape Genes in Merkel Cell Polyomavirus–Positive MCC Cell Lines).
Response: Thank you for directing us to the article “Single-Molecule RNA Sequencing Reveals IFNγ-Induced Differential Expression of Immune Escape Genes in Merkel Cell Polyomavirus–Positive MCC Cell Lines” (Front Microbiol, 2021; doi:10.3389/fmicb.2021.785662). We have revised the Introduction to better reflect the findings of this study, which demonstrate that IFNγ can induce immune escape–related gene expression in MCC cells, highlighting its potential dual role in tumor immunity and resistance.

Reviewer 3 Report
Comments and Suggestions for Authors
Song et al. present an interesting review of the potential role of IFNγ in the pathogenesis and treatment of Mercel cell carcinoma (MCC). The rationale for the review is to raise awareness of a new paradigm wherein IFNγ administration or modulation of the pathways responsible for its elaboration can be coupled with treatement with surgery, radiation, and ultimately chemotherapeutic and/or immunological treatments of MCC.
In brief, the authors present a detailed review of the development of MCC, citing mutation from environmental radiation (e.g., UV radiation) and exposure to DNA viruses as inciting events for the development of MCC. The section outlining the development of neoplasia is well done.
In the following sections, the utility of IFNγ as an antitumor or tumor evasion modulator is presented. The role of IFNγ as either anti or pro tumor growth is dependent on the tumor microenvironment and which cell type (host or tumor) is predominately expressed.
Finally, it is proposed that if specific cellular pathways are targeted to diminish the tumor evasion function of IFNγ, then this molecule and similar molecules produced by its pathway can serve to improve the clinical outcome of patients afflicted with MCC.
I found the work informative and an interesting read. However, there are a number of issues to be addressed.
- Acronym definitions. From the beginning, the text has acronyms presented without definition. For example, on page 2, line 54, ICI is presented as a therapy without a definition. There are several other instances in the text, and the authors would serve the readership if they meticulously made sure that all acronyms are defined.
- Expand the abbreviation list. The author mention only a small number of the numerous abbreviations found in the text and in the figures. Further, they should consider including a few definitions of the molecular entity cited as indicated.
- Figures. The figures are detailed and helpful, but the text is far to small to read in print form. I had to increase the PDF to 150% to be able to read the words within all the figures. These figures need to be expanded to fill the width of each page, not just the usual right-sided column.
- Methods to block the tumor evasion effects of IFNγ. The authors present very little about how the TME containing MCC can be modified with agents that block relevant pathways so that IFNγ may exert primarily antitumor effects in combination with immunological therapy or chemotherapy. I would like to see this section expanded a bit more if possible. Also, a diagram that demarks the specific pathways and sites within the pathways that should be blocked to enhance the antitumor effects of IFNγ would be helpful.
Author Response
Comment 1: Acronym definitions. From the beginning, the text has acronyms presented without definition. For example, on page 2, line 54, ICI is presented as a therapy without a definition. There are several other instances in the text, and the authors would serve the readership if they meticulously made sure that all acronyms are defined. Expand the abbreviation list. The author mention only a small number of the numerous abbreviations found in the text and in the figures. Further, they should consider including a few definitions of the molecular entity cited as indicated.
Response: We sincerely thank the reviewer for this important suggestion. We have carefully reviewed the manuscript and ensured that all acronyms are fully defined upon their first occurrence, including in the abstract—for example, “ICI” is now defined as “immune checkpoint inhibitor.” Additionally, we have substantially expanded the abbreviation list at the end of the manuscript to cover all relevant terms and molecular entities such as MCPyV, TME, CTL, NK, HDAC, APOBEC3B, STAT1/3/5, IFNGR, and IDO1. Where appropriate, brief explanatory notes were added to facilitate reader comprehension. We believe these revisions significantly improve the clarity and usability of the manuscript.
Comment 2: Figures. The figures are detailed and helpful, but the text is far to small to read in print form. I had to increase the PDF to 150% to be able to read the words within all the figures. These figures need to be expanded to fill the width of each page, not just the usual right-sided column.
Response: Thank you very much for your constructive comment. We appreciate your positive feedback on the content of the figures. In response to your suggestion, we have adjusted all figures. We hope these revisions significantly improve the clarity and visual accessibility of the figures.
Comment 3: Methods to block the tumor evasion effects of IFNγ. The authors present very little about how the TME containing MCC can be modified with agents that block relevant pathways so that IFNγ may exert primarily antitumor effects in combination with immunological therapy or chemotherapy. I would like to see this section expanded a bit more if possible. Also, a diagram that demarks the specific pathways and sites within the pathways that should be blocked to enhance the antitumor effects of IFNγ would be helpful.
Response: In response to the reviewer’s request for an expanded discussion on methods to block the tumor evasion effects of IFN-γ, we have substantially revised Section 5 of the manuscript to include more detailed descriptions of relevant therapeutic strategies. Specifically, we now discuss the potential roles of IDO1 inhibitors, epigenetic modulators, and STAT3 inhibitors in mitigating IFN-γ–induced immune suppression and enhancing antitumor efficacy.
Regarding the suggestion to include a diagram delineating specific pathways and molecular targets, we fully appreciate the value of visual summaries. However, as Figure 4 already illustrates the key IFN-γ signaling axis—including PD-L1 and IDO1 upregulation—we opted not to include an additional figure to avoid redundancy and maintain visual clarity. Instead, we have provided a more comprehensive textual explanation to outline potential intervention points and conceptual frameworks for combination strategies involving IFN-γ–based immunotherapy.

Round 2
Reviewer 3 Report
Comments and Suggestions for Authors
No further comments.